# Regulated bacterial interaction networks: A mathematical framework to describe competitive growth under inclusion of metabolite cross-feeding

**Isaline Guex**[1], **Christian Mazza**[1]*, **Manupriyam Dubey**[2], **Maxime Batsch**[2], **Renyi Li**[2], **Jan Roelof van der Meer**[2]*

**1** Department of Mathematics, University of Fribourg, Fribourg, Switzerland, **2** Department of Fundamental Microbiology, University of Lausanne, Lausanne, Switzerland

* christian.mazza@unifr.ch (CM); janroelof.vandermeer@unil.ch (JRVDM)

**Data Availability Statement:** The authors confirm that all data underlying the findings are fully available without restriction. Scripts used for the

## Abstract

When bacterial species with the same resource preferences share the same growth environment, it is commonly believed that direct competition will arise. A large variety of competition and more general 'interaction' models have been formulated, but what is currently lacking are models that link monoculture growth kinetics and community growth under inclusion of emerging biological interactions, such as metabolite cross-feeding. In order to understand and mathematically describe the nature of potential cross-feeding interactions, we design experiments where two bacterial species *Pseudomonas putida* and *Pseudomonas veronii* grow in liquid medium either in mono- or as co-culture in a resource-limited environment. We measure population growth under single substrate competition or with double species-specific substrates (substrate 'indifference'), and starting from varying cell ratios of either species. Using experimental data as input, we first consider a mean-field model of resource-based competition, which captures well the empirically observed growth rates for monocultures, but fails to correctly predict growth rates in co-culture mixtures, in particular for skewed starting species ratios. Based on this, we extend the model by cross-feeding interactions where the consumption of substrate by one consumer produces metabolites that in turn are resources for the other consumer, thus leading to positive feedback in the species system. Two different cross-feeding options were considered, which either lead to constant metabolite cross-feeding, or to a regulated form, where metabolite utilization is activated with rates according to either a threshold or a Hill function, dependent on metabolite concentration. Both mathematical proof and experimental data indicate regulated cross-feeding to be the preferred model to constant metabolite utilization, with best co-culture growth predictions in case of high Hill coefficients, close to binary (on/off) activation states. This suggests that species use the appearing metabolite concentrations only when they are becoming high enough; possibly as a consequence of their lower energetic content than the primary substrate. Metabolite sharing was particularly relevant at unbalanced starting cell ratios, causing the minority partner to proliferate more than expected from the competitive substrate because of metabolite release from the majority partner. This effect thus likely

models in this work can be accessed from https://github.com/IsalineLucille22/Liquid-Model-2.git.

**Funding:** This work was supported by the Swiss National Science Foundation (Sinergia program, grant CRSII5 189919/1) to CM and JvdM, SystemsX.ch grant 2013/158 (Design and Systems Biology of Functional Microbial Landscapes "MicroScapesX") to JvdM, and by the National Centre in Competence Research in Microbiomes (funded by the Swiss National Science Foundation, grant number 180575) to JvdM. The funders had no role in study design, data collection and analysis, decision to publish, or preparation of the manuscript.

**Competing interests:** The authors have declared that no competing interests exist.

quells immediate substrate competition and may be important in natural communities with typical very skewed relative taxa abundances and slower-growing taxa. In conclusion, the regulated bacterial interaction network correctly describes species substrate growth reactions in mixtures with few kinetic parameters that can be obtained from monoculture growth experiments.

## Author summary

Correctly predicting growth of communities of diverse bacterial taxa remains a challenge, because of the very different growth properties of individual members and their myriads of interactions that can influence growth. Here, we tried to improve and empirically validate mathematical models that combine theory of bacterial growth kinetics (i.e., Monod models) with mathematical definition of interaction parameters. We focused in particular on common cases of shared primary substrates (i.e., competition) and independent substrates (i.e., indifference) in an experimental system consisting of one fast-growing and one slower growing Pseudomonas species. Growth kinetic parameters derived from monoculture experiments included in a Monod-type consumer-resource model explained some 75% of biomass formation of either species in co-culture, but underestimated the observed growth improvement when either of the species started as a minority compared to the other. This suggested an important role of cross-feeding, whereby released metabolites from one of the partners are utilized by the other. Inclusion of cross-feeding feedback in the two-species Monod growth model largely explained empirical data at all species-starting ratios, in particular when cross-feeding is activated in almost binary manner as a function of metabolite concentration. Our results also indicate the importance of cross-feeding for minority taxa, which can explain their survival despite being poorly competitive.

## 1 Introduction

Bacteria and other microorganisms colonize practically any habitat on our planet, be it host-associated or in free-living environments [1]. Typically, they occupy their habitats as multi-species communities, that develop as a consequence of their seeding history and migration priorities (i.e., which species entered into the habitat at which moment) [2–4], and under the physico-chemical conditions prevailing in the habitat [5–7]. Multi-species communities intrinsically develop interspecific interactions [8–10], arising from the spatial constellations of cells from different taxa [11, 12], their physiological and metabolic properties [13, 14], and distance- or contact-dependent biological mechanisms [15, 16]. The large physiological and metabolic flexibility and genotype richness make it difficult to determine the main factors underlying community development, and consequently, to capture those factors in appropriate predictive mechanistic models.

A large variety of community models has been put forward over past years, which emphasize either population development within communities from Lotka-Volterra-type interactions, or are based on McArthur consumer-resource theory [17]. Models can be deterministic and population-based [18–20], or cell-oriented [21, 22] and including spatial or flux components [23–25]. Of major importance for the Lotka-Volterra models is the estimation of pairwise interaction coefficients, the true nature or 'strength' of which is mostly not known, but

can be estimated from model fitting. For example, field observations of plant or animal species distributions have been used to infer spatial correlations [26]. In addition, species interactions have been modelled from interacting patches [27–29] or scale transitions [26, 30–32]. In case of microbial communities, interactions are frequently inferred indirectly from longitudinal measurements of relative taxa abundances within community samples, or are parameterised from comparison of single versus paired taxa growth [19, 22, 33, 34]. The general types of interactions occurring between community members can be described from their (positive, negative or neutral) signs at steady state [35]. Lotka-Volterra models typically do not take into account substrate utilization, which governs microbial cell physiology and growth, nor the effects of the environmental or host boundary conditions. Furthermore, they rarely include system feedbacks or multi-species interactions.

In contrast to Lotka-Volterra models, which mostly function as a 'top-down' approach on experimental or field observations, agent-based models consider microbial cells as mathematical objects (with or without physical size), whose division and population growth is controlled by local substrate concentrations and physico-chemical conditions [23–25]. The spatial environment is broken down into grids, for which the local substrate and conditions can be calculated, and onto which the agents are placed. Agents are given metabolism and cell division properties (i.e., Monod kinetics) [36], or other biological properties such as motility or chemotaxis (i.e., Patlak-Keller-Segel model) [37]. Depending on local conditions, the agents will deplete substrate, divide and change the grid concentrations, and will occupy different positions on the surface. Concentrations and agent positions are calculated for every time step, from which the species growth or movement on the surface can be quantified. Species interactions can be introduced by defining interaction parameters that influence Monod kinetics on the substrate [36]. Agents can be given more complex metabolic functioning by including individual reduced genome-scale metabolic models [38]. This enables modeling of substrate uptake, and efflux and exchange of metabolites between species. Additional complexity, for example, from cell crowding, can be included to more realistically represent multicellular structures on surfaces. The individual based models have been further adapted to 3D porous environments, to include species distributions in space [39].

Developing a good mathematical model to describe bacterial species interactions is crucial for community growth predictions, but depends on the level of intended detail and granularity in the system (e.g., system- or agent-based), the specific questions being asked and the types of available data. Particularly for microbe-centered community models it would be important to connect to classical theory of microbial growth kinetics and physiology, such that empirical measurements of the latter (e.g., growth rates, yields) can more easily be included, but such models have not been widely used [40]. Furthermore, there is no consensus yet on how to mathematically best link growth kinetic models of pure cultures and that of multi-species mixtures, under inclusion of emerging biological and metabolic interactions [17]. The primary objective of the work presented here was thus to describe a mathematical basis for resource-limited community growth under the influence of interspecific interactions. On the basis of forward regulatory feedback theory, we propose a generalized interaction parameter, which connects to the individual species' maximum specific growth rate. Our basic hypothesis is that population growth rates within bacterial communities are mostly influenced by carbon and nutrient availability (and less so by biological warfare mechanisms between species), and thus strongly dependent on consumption of primary substrate(s) in the system, as well as on production and utilization of metabolites (byproducts) from the primary substrates (see, for example, Refs. [41] [40]).

We develop the model notions for the case of a two-species system, assuming that once such interaction terms are defined and parameterized, it will be easier to extend their

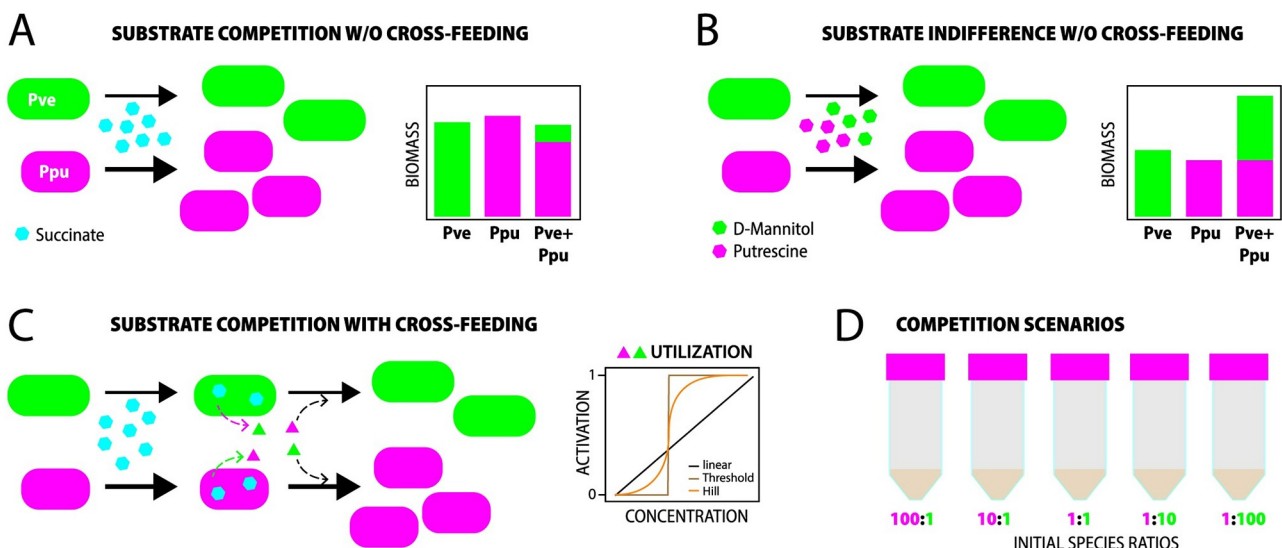

**Fig 1. Conceptual and experimental design of *P. veronii* and *P. putida* co-culture experiments.** (**A**) Simple direct competition for a single primary substrate (succinate, here at 5 mM), leading to expected stationary phase biomass differences in co-culture conditions because of faster growth kinetics of *P. putida* (Ppu, magenta) than *P. veronii* (Pve, green). (**B**) Expected outcome for a scenario of substrate indifference; a single unique substrate for either of the strains. (**C**) Scenario of arising metabolite cross-feeding and the imposed activation type function on reciprocal utilization of the excretion products. (**D**) Empirical co-culture conditions to facilitate detection of yield-effects from cross-feeding.

description and usage to higher order mixtures and conditions (e.g., as in [41]). Our general population growth model is based on utilisation of a single growth-limiting resource in batch culture, which we extend by including metabolic interactions using a chemical reaction network (CRN) [20], which differs from so-called generalized consumer-resource models ([17, 41]). We then introduce metabolite-based cross-feeding where the consumption of a substrate by one of the consumers produces metabolites that can act as resources for the other, leading to positive feedback in the two-species system (Fig 1). We prove mathematically that such positive feedback reaction cannot be constant, since they would lead to vanishing steady state metabolite concentrations. This is in contradiction to previous data (see, e.g., [41] where steady-state waste concentrations were shown to persist) and results in violation of thermodynamic principles. Instead, we introduce the concept of activation thresholds, in analogy to previous gene regulatory network models [42–44], to make the positive feedback conditionally dependent on (summed) metabolite concentrations. Model predictions were tested and compared to experimental resource-limited batch growth with two species, one of which a fast (*Pseudomonas putida*) and the other a slower-growing bacterium (*Pseudomonas veronii*) (Fig 1A). Both species were grown individually or in co-culture in chemically defined medium, using different starting cell ratios and abundances. Population growth rates were inferred from exponential phase, and overall biomass yields from stationary phase measurements. We further experimentally impose two types of substrate scenarios, either using a primary single competitive resource (i.e., succinate, Fig 1A) for both bacterial species, or invoking substrate 'indifference', in the form of two substrates specific for each of the two bacteria (Fig 1B). Data and model simulations indicate that utilization of excreted metabolites is a non-negligible part of competitive interactions. Contrary to intuition, this leads to minority populations proliferating better than expected from growth kinetic differences in competition. This effect may thus help to understand why slow-growing taxa in natural communities are persisting even under substrate competition.

## 2 Results

### 2.1 Development of a regulated community growth interaction model

We use the notions and tools from chemical reaction network theory (CRN; see Materials and methods) to expand community growth kinetic models with interspecific interactions that assume metabolic cross-feeding (Fig 1C). The CRN for a single species and substrate, under formation of an intermediary complex, resolved to new cell biomass leads to a mathematical description that is identical to the classical empirical Monod equation (Appendix B in S1 Text, Eq. 18). Similar to other previous approaches, we assume that during metabolism and biosynthesis, substances other than cell biomass are produced [40, 45–49], which can leak outside the cell. Leaking substances may consist of regular metabolites in temporary overflow (which the cell may take up again at a later stage), or specifically excreted compounds with a biological distinct function (e.g., signalling molecules or toxins), or wastes (i.e., compounds not further used by the species). In contrast to other models (and in absence of detailed information on the actual excreted metabolites in the system) we avoid the fine-grain complexity of estimating uptake rates of individual specific known metabolites and formation of biomass building blocks (as assumed in Ref [40]), but propose here a simplified lumped waste ($W$). The advantage of a lumped waste is the ease to model a regulated feedback for the two-species system, as will be explained below, and calculate a global estimate as to how much the waste contributes to yield gains or losses of either species in competition.

Growth of either species $S_1$ and $S_2$ in co-culture with a single common resource ($R$) can be generally conceived as a CRN of species-substrate reaction with newly produced species under concomitant production of species-specific wastes $W_1$ and $W_2$ (Eq 1).

$$S_1 + R \xrightarrow{\kappa_{1_1}} P_1 \xrightarrow{\kappa_{1_2}} 2S_1, \quad P_1 \xrightarrow{\kappa_{1_3}} S_1 + W_1,$$
$$S_2 + R \xrightarrow{\kappa_{2_1}} P_2 \xrightarrow{\kappa_{2_2}} 2S_2, \quad P_2 \xrightarrow{\kappa_{2_3}} S_2 + W_2. \tag{1}$$

Both species, denoted by $i$ = 1, 2, will take up the resource with a rate $\kappa_{i_1} \in \mathbf{R}^+$, producing an intermediary complex $P_i$, which with a rate $\kappa_{i_2} \in \mathbf{R}^+$ can either divide and/or, with a rate $\kappa_{i_3} \in \mathbf{R}^+$, produce a species-specific waste $W_i$. In the following, we omit the label of the species, $i$, when generally speaking about both species. Importantly, in the CRN of Eq 1 the yield outcomes of both species in co-culture are determined by the inherent kinetic properties of either species (e.g., as in Fig 1A), which can be estimated from the respective monoculture growth profiles.

We can now extend this scenario of primary resource competition to one where the species not only grow on the primary resource $R$ and produce species-specific wastes $W_1$ and $W_2$, but can also utilize the produced waste. In case of a reciprocal waste utilization, this leads to competitive cross-feeding in the system (Fig 1C, Appendices S3 and S6 in S1 Text). Similar situations have been considered in Refs. [17, 40, 41] for generalized consumer-resource models, and also in [20] in relation to Lotka-Volterra pairwise interaction models (S1 Fig). Obviously, species-specific wastes may also be re-utilized by the species in question (for example, in case of temporarily excreted metabolic intermediates, see, e.g., Ref [45]), but this is implicit in yield measurements of monoculture growth and is therefore not considered here.

We can assume metabolites to be produced by growing and dividing cells as a function of dynamic fluxes in catabolic and anabolic reactions, and some of those may be excreted to the outside as a result of leakage, balancing energy overflow or improperly expressed pathways [45–49]. However, depending on the growth environment and the concentration of producer cells, such metabolite concentrations may be too low to be taken up by the cross-feeding

species [36]. In particular, at the onset of batch growth with low starting cell numbers, high substrate concentrations and a large medium volume, the appearing waste concentrations will be immediately diluted by molecular diffusion. To account for the fact that metabolite concentrations are too low for efficient uptake by cross-feeding cells, and have less energy content than the primary substrate, we introduce a concentration-dependent threshold function for the utilization of the waste (Fig 1C). At the start of growth, both species preferentially only consume the primary resource because of its higher concentration, and it will take some time for the waste to appear in the system before being detected and used by the (respective) other species.

We can take this effect into account by constraining the utilization of waste as growth resource by a threshold function (with species-specific thresholds $T_{W_1}$ and $T_{W_2}$), that follow an activation function $\phi(x)$ with $\phi(x) \approx 0$ when $x = < 0$ and $\phi(x) \approx 1$ when $x > 0$, which is dependent on the differences between the concentrations $F_i$ of wastes $W_i$ and their thresholds $T_{W_i}$.

In the following, we simulated the effects of two types of threshold functions. In the first case, the threshold function is discontinuous, whereas in the second, waste is utilized as a continuous function of its concentration. In the first case (discontinuous threshold) we take $\phi(x) = \mathbb{1}_{\{x>0\}}$ which is 1 when $x > 0$ and zero otherwise. When implemented in the mass action o.d.e., the resulting mass action term contains factors $\mathbb{1}_{\{F_i \geq T_{W_i}\}}$ that force reactions to occur only when the concentration of $W_i$ exceeds the threshold $T_{W_i}$. Biologically speaking, this would mean that either of the species $S_1$ and $S_2$ is able to take up the excretion products of the other as soon as its concentration is above the threshold.

In the second case, we choose $\phi$ as a continuous sigmoidal activation function with $0 \leq \phi(x) \leq 1$, such as is generally described by the Hill function (S2 Fig). Activation of feedbacks through steep sigmoidal functions are regularly used in description of gene regulatory networks to describe transcriptional activation or repression as a function of transcription factor and effector concentrations (see e.g. [50]). Similarly, uptake of the primary resource $R$ and utilisable waste molecules $W$ can be described by physical contacts and binding of the substances to transporters at the cell surface, justifying the use of a Hill activation function. Growth of the two species in co-culture under inclusion of cross-feeding by appearing waste for a general activation function $\phi$, is then described by the following CRN (simplified by eliminating the intermediate species $P_1$ and $P_2$ (see [42]). We call this a regulated bacterial network (RBN, Appendix F in S1 Text).

$$S_1 + R \xrightarrow{\kappa_1} 2S_1, \quad S_1 + R \xrightarrow{\tilde{\kappa}_1} S_1 + W_1,$$
$$S_2 + R \xrightarrow{\kappa_2} 2S_2, \quad S_2 + R \xrightarrow{\tilde{\kappa}_2} S_2 + W_2, \tag{2}$$

$$S_1 + W_2 \xrightarrow{\phi\left(F_2 - T_{W_2}\right)\kappa_{21}} 2S_1, \quad S_1 + W_2 \xrightarrow{\phi\left(F_2 - T_{W_2}\right)\tilde{\kappa}_{21}} S_1 + W_1,$$
$$S_2 + W_1 \xrightarrow{\phi\left(F_1 - T_{W_1}\right)\kappa_{12}} 2S_2, \quad S_2 + W_1 \xrightarrow{\phi\left(F_1 - T_{W_1}\right)\tilde{\kappa}_{12}} S_2 + W_2, \tag{3}$$

with all rates $\kappa$ and $\tilde{\kappa}$ being constant real positive numbers.

This RBN is different from previous consumer-resource models, which allow permanent cross-feeding (which corresponds to the situation where $\phi \equiv 1$). However, allowing permanent cross-feeding will lead to vanishing steady-state waste concentrations (Appendix D in S1 Text; Propositions 1 and 2), which is in contradiction to experimental results from literature (steady-state waste concentrations do not vanish, see, e.g. [41]). The concentration-dependent

threshold function on waste utilization overcomes this, and recapitulates the experimental results described below. The mathematical deduction of the RBN into dynamic o.d.e. which simulate the corresponding biomass growth of either species over time as the result of primary substrate and/or reciprocal waste utilization is presented in S1 Text (Appendices S5 and S6).

## 2.2 Co-culture growth prediction from monoculture-derived kinetic parameters without cross-feeding

We experimentally use a tractable system of two fluorescently labeled bacterial strains *P. veronii* and *P. putida* to test and verify the scenarios of primary resource competition, under inclusion or not of competitive metabolite cross-feeding (Fig 1). Both strains were hereto grown in liquid suspension, either as monoculture or as a co-culture. We imposed different starting ratios and/or total abundances of both strains, anticipating that yield-effects of metabolite cross-feeding may only be detected under very skewed starting ratios (Fig 1D). Population growth and relative strain abundances were measured from increases in culture turbidity, from strain-specific fluorescence and by flow cytometry. For direct competitive growth, strains were cultured on succinate as the sole added carbon and energy source (dissolved at 5 mM). By using the direct competitive substrate model without cross-feeding, we first estimated the proportion of co-culture yields, which can be described from the inherent individual growth kinetic properties, as measured from respective monoculture growth.

The relevant growth kinetic parameters extracted from monoculture growth include the maximum growth rate ($\mu_{max}^{(i)}$), the half velocity constant ($K_S^{(i)}$) and the biomass yield ($\rho_i$). These parameters can be expressed as functions of the reaction rates ($\kappa_1$, $\kappa_2$ and $\kappa_3$) (see Eq 1, and S1 Text, Appendix B to have their explicit form).

First, we estimated the density distributions of the kinetic parameter values from half of the data sets (n = 5) using the Metropolis-Hasting algorithm with a Markov Chain Monte Carlo approach (Fig 2A and B, Appendix H in S1 Text), and converted these to biomass values using separate empirical cell dry weight estimations (Materials and methods). Growth rates of *P. veronii* were 2.2 times slower, and its fitted yields were 75% of those of *P. putida* (Fig 2A). In contrast, the value fitted for the $K_S$ of *P. veronii* for succinate was lower than that of *P. putida*. Simulations of monoculture biomass growth of both *P. putida* and *P. veronii* with the estimated kinetic parameters from half of the replicates, were closely overlapping with the observed increase in biomass in the other five replicate data sets (Fig 2C). The only exception to the simulation was a slight observed decrease in turbidity of stationary phase *P. putida* cultures, which is due to decrease in cell sizes and cell clumping, and which the growth model does not take into account. This indicated that we could use the estimated parameter values to simulate co-culture growth.

Turbidities of co-cultures with 5 mM succinate increased similarly as those of *P. putida* alone (Fig 3A), suggesting that they consisted in the majority of *P. putida*. This was confirmed qualitatively by measurements of the species-specific fluorescent markers during growth, which showed almost similar *P. putida* fluorescence in the co- as in the monoculture, and less than 20% GFP fluorescence from *P. veronii* in the co-culture (Fig 3B). (Since the per cell fluorescence changes as a function of growth phase, the fluorescence plots cannot be converted to species-specific biomass).

Simulated co-culture growth with kinetic parameter estimates from the individual monocultures predicted that the population of *P. veronii* indeed represents a minority in the co-culture (Fig 3C). Prediction of summed co-culture biomass followed closely the observed co-culture growth, but with the Monod co-culture model giving a better fit than a logistic model, which tended to deviate in the early exponential and stationary phase (Fig 3D; Appendix A in

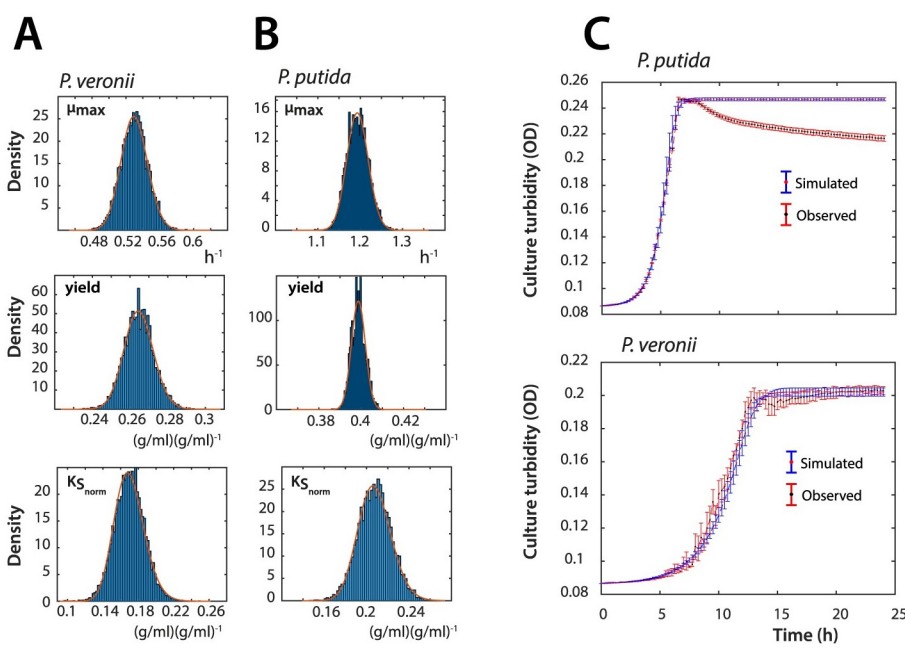

**Fig 2. Extracted growth kinetic parameters and simulated monoculture growth of *P. veronii* and *P. putida*. (A)**
Estimations of $\mu_{max}$, $\rho$ (yield) and $K_S$ for *P. veronii*. Diagrams show histogram (blue bars) and log-normal (red lines)
distributions of parameter estimations from 5 out of 10 biological replicates of *P. veronii* grown in mixed liquid
suspension on 5 mM succinate, by using the Metropolis-Hasting algorithm. Note that $K_S^{(i)}$ is normalized by dividing by
the initial biomass in *g*. (**B**) as A, but for *P. putida*. (**C**) Simulated (blue) versus observed monoculture growth (red) of
either *P. veronii* or *P. putida*, using the parameter sets of A and B. Observed growth is plotted as mean culture
turbidity ± one *sd* from the 5 replicate cultures not used for parameter fitting. Culture turbidity transformed from
calculated biomass by measured conversion factors in g cell dry weight per ml OD culture (see Materials and
methods). S1 Data.

S1 Text). As expected, both models did not capture the observed decrease in co-culture turbidity, because neither of them includes any process that describes cell size decrease or clumping.
Both models predicted stationary phase proportions of *P. veronii* biomass of approximately 3%
instead of the 4% measured empirically by flow cytometry in the co-cultures (Fig 3E), n = 10
simulations and n = 3 biological replicates, p-value = 0.0425 using two-sided t-test). This small
but statistically significant difference would indicate that the bulk part (75%) of the growth
behaviour of both species in co-culture under primary substrate competition is determined by
their inherent kinetic physiological differences, whereas 25% must be due to interactions
between *P. veronii* and *P. putida*.

## 2.3 Consistent deviation from predicted competitive behaviour at imbalanced starting cell mixtures

The results obtained above indicated that the inherent growth kinetic distinctions between
two species can quickly result in large yield differences, which make it more difficult to empirically quantify cross-feeding effects. In order to more systematically test this, we repeated co-culturing at different starting cell densities and ratios of *P. veronii* to *P. putida*, again using 5
mM succinate as the sole added carbon and energy source for the cells. Model simulations
(without cross-feeding) over a range of starting cell ratios (0.1–0.999 of *P. veronii* to *P. putida*,
Fig 4A) and/or densities (from $10^5$–$10^7$ cells per ml) suggested that, purely from inherent
growth kinetics, *P. putida* will dominate the co-cultures in steady-states, except with 10–

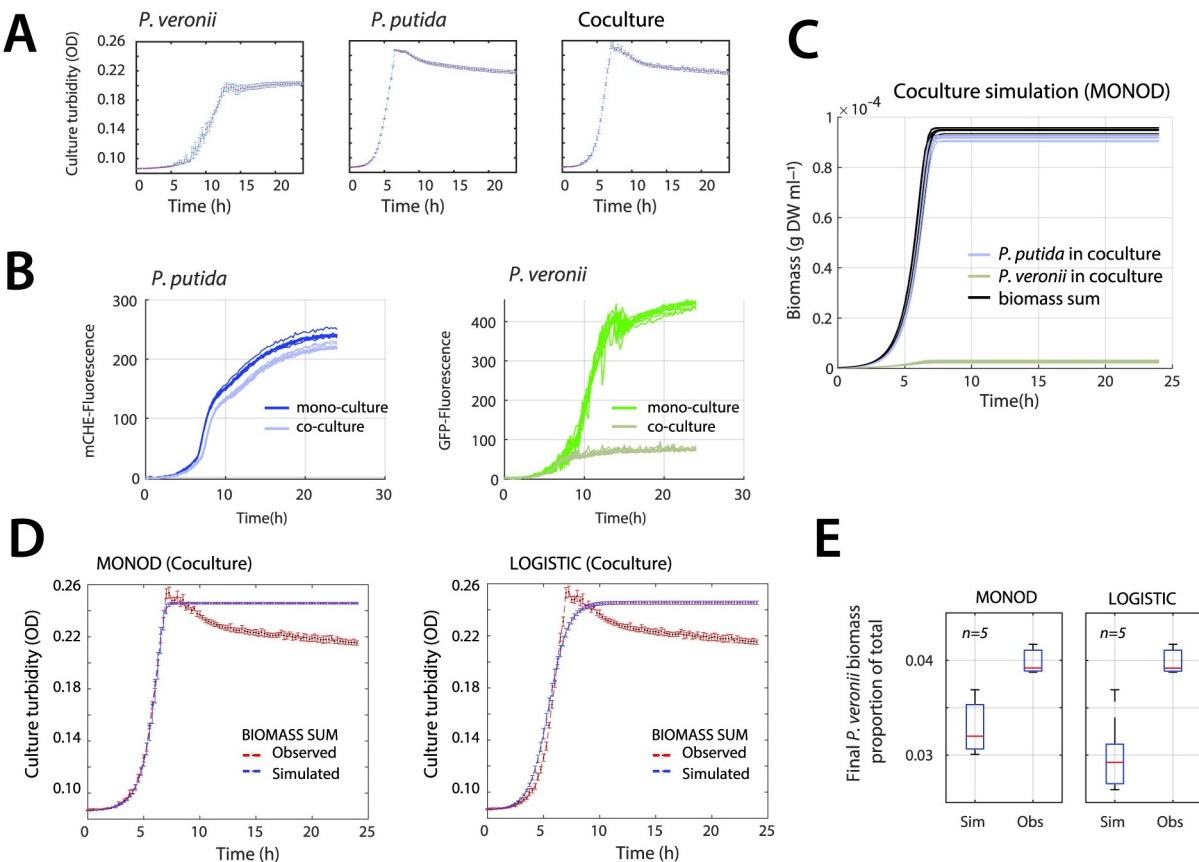

**Fig 3. Observed and simulated growth of *P. veronii* and *P. putida* co-cultures under substrate competition.** (**A**) Experimental observations of growth on 5 mM succinate (as culture turbidity). Data points indicate the mean culture density (red dots) ± the calculated standard deviations (vertical error bars, *n* = 10 replicates). The co-culture starts as a 1:1 mixture of both species' cell numbers. (**B**) Species-specific fluorescence in mono- and co-cultures. *P. putida* biomass is recorded in the mCherry and *P. veronii* in the GFP channel. Lines show individual replicates (*n* = 10). (**C**) Simulations (*n* = 10) of individual species' growth and summed biomass in 1:1 starting co-cultures on 5 mM succinate as sole carbon substrate, using the Monod co-culture model and kinetic parameter sets from monoculture estimations, without any cross-feeding interactions. (**D**) Comparison of Monod and logistic co-culture models for simulating the total co-culture biomass growth on 5 mM succinate. (**E**) Final proportion of *P. veronii* of the total biomass for both co-culture models at steady-state (experimental observations use the species-specific cell counts from flow cytometry transformed into species-specific biomass as g DW ml$^{-1}$). S2 Data.

100-fold surplus of *P. veronii* and high cell starting densities (Fig 4B). At higher starting cell densities, the availability of resources permits fewer generations of growth, as a result of which *P. putida* cannot gain that much advantage as it would at low starting cell densities (Fig 4B). By comparing to the actual measured species proportions under those conditions, we see that kinetic predictions (i.e., without cross-feeding) are in agreement to empirical outcomes only when species ratios are equal at start (1:1, Fig 4C). In contrast, at imbalanced starting ratios, either of the two species benefits from the presence of the other in improving its growth. For example, at starting ratios of *P. veronii* to *P. putida* of 1:100 and 1:10, the observed final *P. veronii* proportion is statistically significantly higher than expected from its derived monoculture kinetic parameters, whereas at ratios of 100:1 and 10:1, its final proportion is lower than expected (and *P. putida* proportions are higher; Fig 4C). This confirmed that the species are likely to engage in some other interaction than only direct competition for resource, as a result of which—in particular, relatively small starting abundances of one species can profit from a larger abundance of the second species.

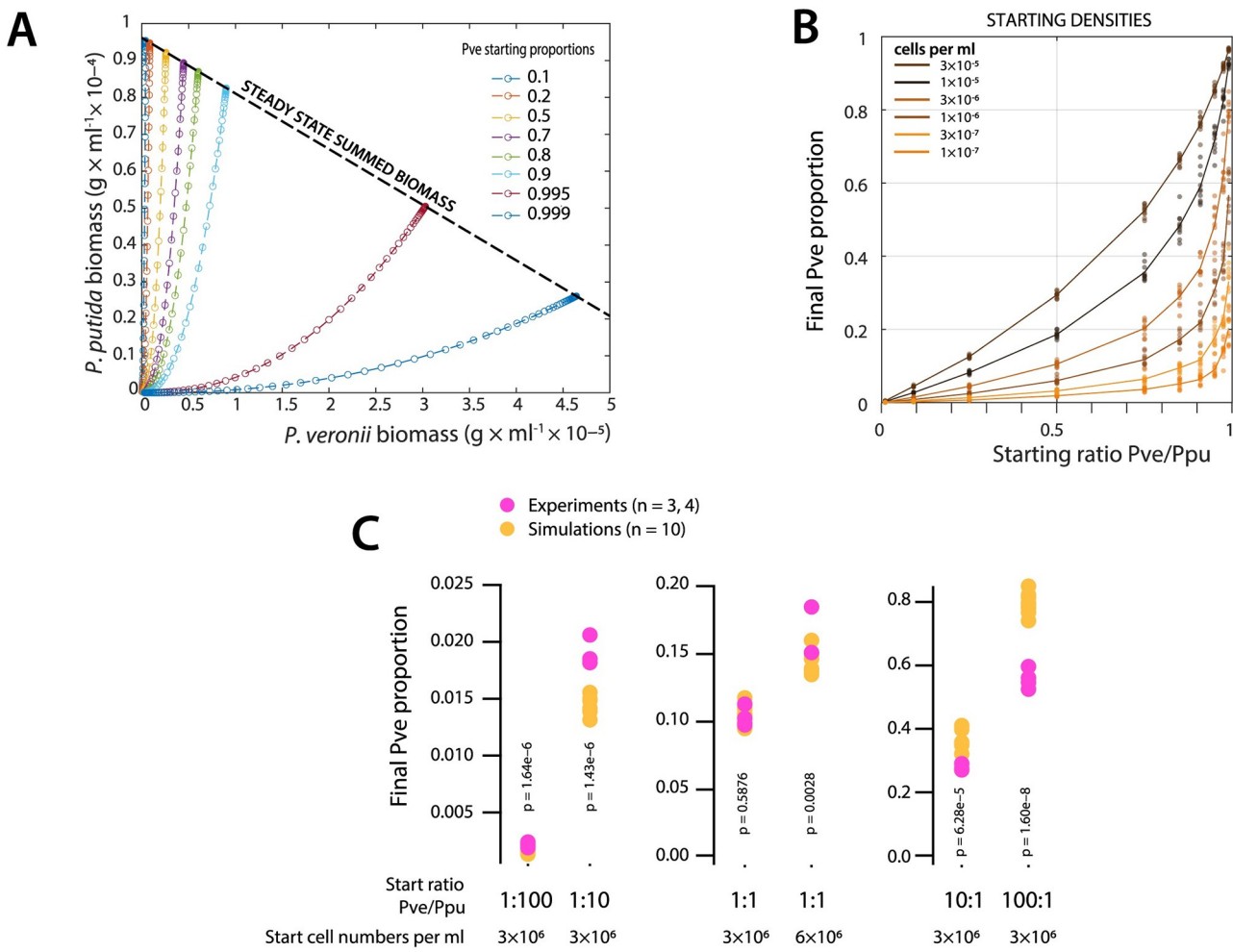

**Fig 4. Effect of initial biomass concentration and starting cell ratios in a two-bacterial system growing on a single substrate on their steady-state distributions.** (**A**) Biomass of *P. veronii* (Pve) and *P. putida* (Ppu) given in g DW ml $^{-1}$ with an initial resource concentration of 5 mM succinate (equivalent to $2.4 \times 10^{-4}$ g ml$^{-1}$). Simulated growth trajectories (Monod model without cross-feeding) for the different species at their indicated starting proportions, shown as coloured lines and symbols according to the legend. (**B**) Steady-state biomass concentrations as a function of different starting cell ratios and starting cell densities (colours, in g biomass DW ml $^{-1}$). Symbols correspond to individual replicate simulations, with log-random sampled kinetic parameters from estimated distributions of Fig 2. (**C**) Difference of observed (magenta) versus simulated (ochre) final *P. veronii* proportions at different initial ratios (bottom) and two starting densities. p-values stem from paired two-sided t-test comparisons. S3 Data.

## 2.4 Cross-feeding models explain utilization of part of excreted metabolites at imbalanced starting cell ratios

To estimate the potential effect resulting from cross-feeding interactions as we theorised conceptually above in the RBN species reaction model, we simulated the production of waste (*W*) from the fitted $\kappa_1$, $\kappa_2$ and $\kappa_3$ values for each of the species in monoculture. In absence of cross-feeding (Fig 5A), one can see that both species convert a significant quantity (60%) of the primary substrate succinate (5 mM, equivalent to $2.5 \times 10^{-4}$ g C ml $^{-1}$) into waste (here taken as the sum of all product not incorporated into biomass, including $CO_2$). This is not surprising, as excretion of up to 50% of re-usable carbon metabolites (apart from released $CO_2$) has been routinely described to occur from individual strains growing in batch culture with large primary substrate quantities (i.e., mM-range) [41, 48]. By allowing cross-feeding to occur, one can observe that, depending on the cross-feeding model, the appearance of waste in the co-

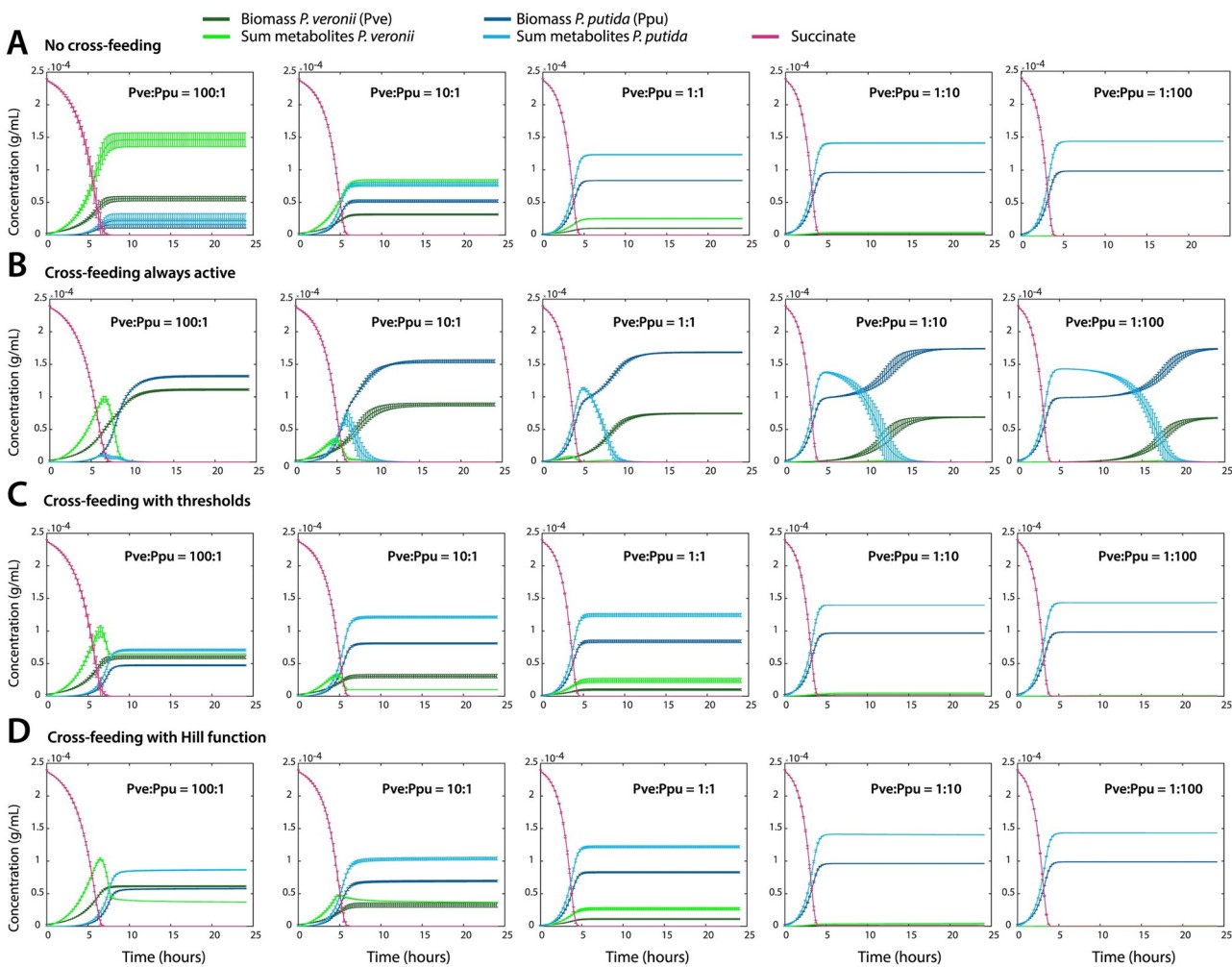

**Fig 5. Simulated biomass growth and waste formation in co-culture, with or without metabolite cross-feeding. A-D**. Biomass growth of either species *P. putida* (dark green) or *P. veronii* (dark blue) on a single shared carbon substrate (5 mM succinate), and predicted waste concentrations (light green and blue), for five different cell starting ratios (100:1, 10:1, 1:1, 1:10 and 1:100, as indicated), and $1 \times 10^6$ cells ml$^{-1}$ at start. Simulations in (A) assume no cross-feeding, those in (B) assume permanent cross-feeding (i.e., $\phi \equiv 1$). Simulations in (C) estimate cross-feeding with the discontinuous threshold (with threshold values (in g mL$^{-1}$), from left to right, $T_{W_1} = 6.5 \times 10^{-5}$, $1.0 \times 10^{-5}$, $2.8 \times 10^{-5}$, $2.8 \times 10^{-5}$, and $2.8 \times 10^{-5}$; $T_{W_2} = 1.4 \times 10^{-4}$, $1.4 \times 10^{-4}$, $1.4 \times 10^{-4}$, $1.4 \times 10^{-4}$, and $1.43 \times 10^{-4}$). The simulations in (D) assume cross-feeding following a Hill activation function with parameter values, $T_{W_1} = 4.73 \times 10^{-5}$ g mL$^{-1}$, $T_{W_2} = 1.73 \times 10^{-4}$ g mL$^{-1}$ and $k = 34$. $T_{W_1}$ corresponds to the threshold of the waste produced by *P. veronii* that is used by *P. putida*; and $T_{W_2}$ vice versa. Plots show the means (lines) from *n = 10* simulations, with error bars representing one standard deviation. Variation is introduced from resampling the distribution of kinetic parameters, as in Fig 2. S4 Data.

culture is diminished and profiting for biomass growth of the reciprocal species (as expected; Fig 5B–5D). This simulation already visually indicates that allowing all waste to be re-utilized by the reciprocal species (i.e., $\phi \equiv 1$ in Eq. 44 of Appendix F in S1 Text) is not in agreement with observed co-culture growth, because it leads to extended growth at later phases (i.e., >10 h; Fig 5B, compare to Fig 3). Interestingly, the models further predict that it is primarily *P. putida* which at low relative starting abundances can profit from waste excreted by the larger thriving population of *P. veronii*, but less so in reverse (Fig 5C and 5D). By comparing the steady-state values of the simulated *P. veronii* waste concentration at the highest starting ratio of Pve:Ppu = 100:1 in absence of cross-feeding and with threshold cross-feeding, we would

obtain a total of $1 \times 10^{-4}$ g C ml $^{-1}$ that is excreted by *P. veronii* and used by *P. putida* for its growth (Fig 5C and 5D). We also considered if the interactions between *P. veronii* and *P. putida* could be driven by inhibition, either reciprocally or unilaterally. Such a scenario, however, results in underestimation of co-culture biomass and largely wrongly simulated proportions of *P. veronii* in the co-culture (Appendix G in S1 Text; S3 Fig), and is therefore, less likely. These simulations thus showed that the co-culture interactions are mostly driven by competitive cross-feeding, and that cross-feeding is most optimally observed at unequal starting cell ratios, when inherent growth kinetic parameters are different, as in the case of *P. putida* and *P. veronii*.

## 2.5 Discontinuous utilization of reciprocal wastes in co-cultures

In order to better understand the effects of discontinuous reciprocal waste utilization, we simulated a range of threshold values on the predicted stationary phase abundances of *P. veronii* and *P. putida* in co-culture, at a starting ratio of 10:1. In case of the discontinuous threshold function, the uptake rates of byproducts are assumed to be equal to that of the primary resource (hence, rates $\kappa_{1_4}$ and $\kappa_{2_4}$ equal to $\kappa_{1_1}$ and $\kappa_{2_1}$, respectively, as in Eq 1, see Appendix F in S1 Text). In this case, only the value of the threshold parameters $T_{W_1}$ and $T_{W_2}$ impacts the predictions of the stationary phase species biomass proportions. As one can see in Fig 6A–6C), varying the threshold values on the utilisation of $W_1$ by *P. putida* and of $W_2$ by *P. veronii* has an important effect on their stationary phase proportions; their closeness to experimental observations and the corresponding utilization of reciprocal waste products. For instance, the steady-state proportions of *P. veronii* are lowest when its threshold for using the byproducts from *P. putida* is highest (i.e., at $T_{W_2} = 0.00015$, Fig 6A), and *P. putida* continuously takes away byproducts from *P. veronii* (i.e., $T_{W_1} = 0$). On the contrary, steady-state proportions of *P. veronii* are highest when *P. putida* has a high threshold on using byproducts from *P. veronii* and *P. veronii* has a low threshold on using byproducts from *P. putida* (i.e., $T_{W_1}$ is 0.000135, and $T_{W_2}$ is 0, Fig 6A). The latter scenario is less likely to be in agreement with experimental data, as plotting the corresponding ratio of simulated versus observed stationary phase species ratios shows (Fig 6B). This suggests that *P. putida* is more efficient in utilizing byproducts from *P. veronii* than the other way around, and, as mentioned, is predicted to profit from up to $1 \times 10^{-4}$ g C ml $^{-1}$ waste product from *P. veronii* (Fig 6C). The physiological reason for this efficiency could lay in the lower $K_m$ of its uptake systems or in the types of metabolites produced and released by *P. veronii*.

Comparison across different simulations (inducing variations from subsampled kinetic parameters), the five starting ratios and the three models, showed that including a threshold on metabolite cross-feeding gives the better explanation of the observed steady-state *P. veronii* proportions in co-culture with *P. putida* on 5 mM succinate (Fig 6D and 6E). Threshold values for the discontinuous model (M2) were derived from the parameter screens, but are arbitrary fits for each of the starting cell ratios. In contrast, the Hill function parameter values were fitted on the experimental data for the complete data sets (as described in the Materials section; S2 Fig). Using the fitted values, we find a maximal difference of about 13% (Fig 6D; M3) at stationary phase between the measured and simulated abundances. In comparison, the model without cross-feeding predicts a maximum difference of about 30%. This indicates that there is a significant interspecific interaction term that develops during growth on a single shared resource, and from which the lesser abundant species can profit. This is counterintuitive from the concept of competition but makes sense from a microbiological perspective, because an excreted waste by an abundant population contains potentially sufficient carbon to support measurable growth of a small population of a second species, but not of a large one. However,

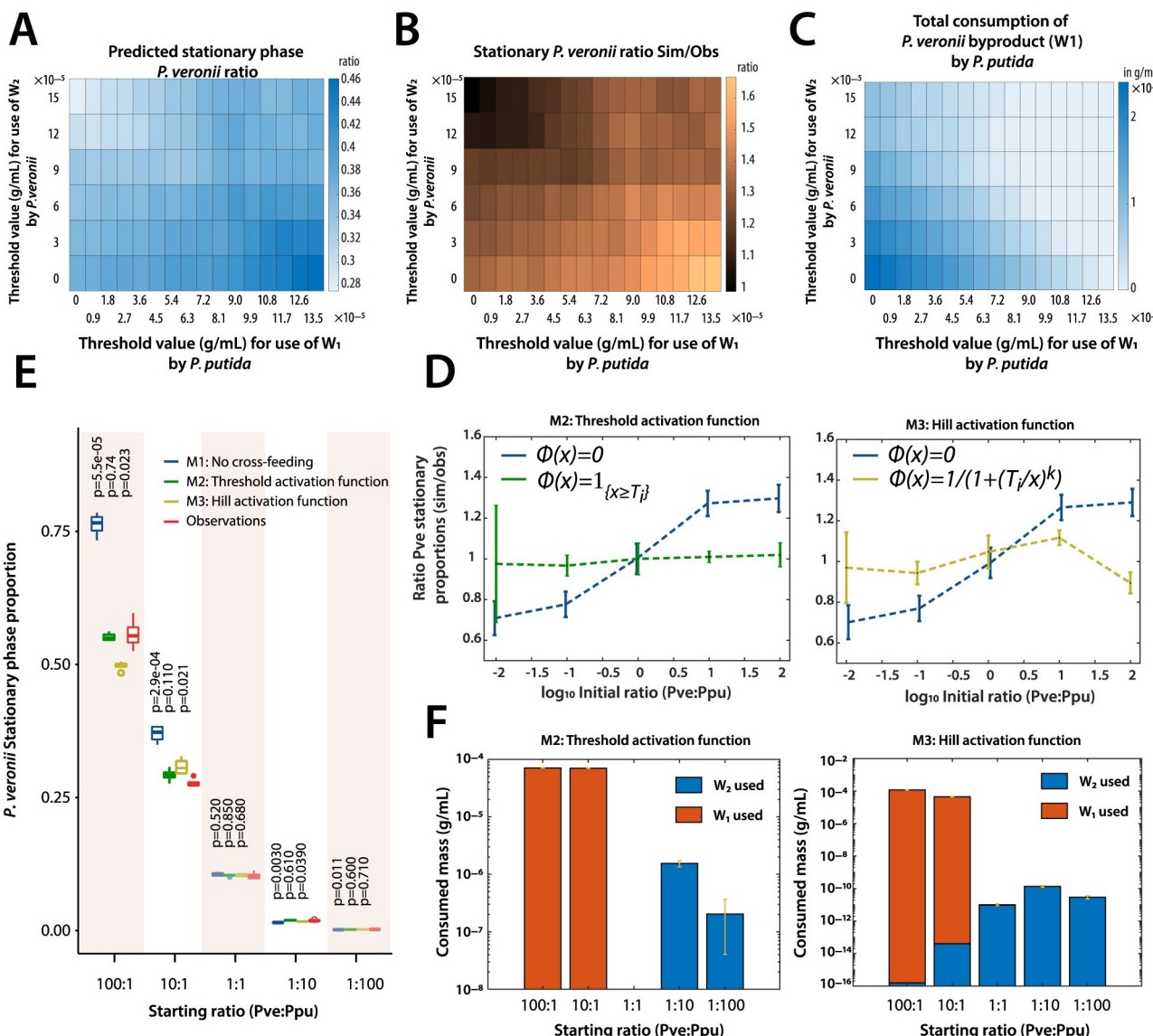

**Fig 6. Effect of threshold values on cross-feeding between both species in co-culture.** (**A**) Predicted stationary phase ratios of *P. veronii* biomass as a function of the discontinuous threshold value for activation of cross-feeding by *P. putida* on the byproducts of *P. veronii* ($W_1$), and vice versa ($W_2$). Ratios plotted as heat map according to the colour legend as indicated. (**B**) Similarity of simulated versus observed stationary phase biomass proportion of *P. veronii*, taken as the ratio; again as function of selected values of $T_{W_1}$ and $T_{W_2}$. (**C**) Predicted waste utilization by *P. putida* as a function of $T_{W_1}$ and $T_{W_2}$. (**D**) Mean relative difference of simulated versus observed stationary phase proportions of *P. veronii* in co-culture with *P. putida* on 5 mM succinate at different starting cell ratios (as indicated 100: 1, 10: 1, 1: 1, 1: 10, and 1: 100). Three model predictions are shown: M1, without cross-feeding (in blue), overlaid with either M2 (model with discontinuous threshold function, in green); or with M3 (Hill activation function, in yellow). (**E**) Comparison of observed steady-state *P. veronii* biomass proportions (observations, in red; *n = 4* replicates) at different starting cell ratios, to those resulting from the three model simulations (M1, M2 and M3; *n = 4* replicates), as above. Box plots show the median, second and third quadrants, and lines indicating the 25th and 75th percentiles. P-values calculated by t-test in comparison to the empirical values. (**F**) Total estimated utilization of waste (in g C mL$^{-1}$) for each of the co-cultures and varying initial cell ratios, as indicated. In blue, consumption of byproducts by *P.veronii*. In red, consumption of byproducts by *P.putida*. For descriptions of the threshold models, see Appendix F in S1 Text. Steady-state cell concentrations of both species were measured by flow cytometry and identified on the basis of the respective fluorescence marker. Cell concentrations were converted to biomass using calculated single cell mass values, as described in the Materials section. Model M2 simulations used threshold values of $T_{W_2} = 1.4 \times 10^{-4}$, and $T_{W_1} = 6.5 \times 10^{-5}$ (ratio 100:1); $1.0 \times 10^{-5}$ (ratio 10:1); and $2.8 \times 10^{-5}$ (all other ratios). Model M3 simulations used the fitted parameters: $T_{W_2} = 1.73 \times 10^{-4}$, $T_{W_1} = 4.73 \times 10^{-5}$, and $k = 34.5$. S5 Data.

both thresholding models on cross-feeding predict that *P. putida* can utilize 2 (model M2) to 6 (model M3) orders of magnitude more of the reciprocal waste than *P. veronii* (Fig 6F).

## 2.6 Co-culture growth with two independent resources

To contrast direct substrate competition with appearing cross-feeding with a situation in which both species would be solely driven by growth kinetic differences, we designed an experimental scenario of substrate 'indifference' (Fig 1B). In the corresponding CRN, each species now has its own resource (e.g., $R_1$ and $R_2$; see Appendix C in S1 Text). We repeated co-culturing of *P. putida* and *P. veronii*, but in this case with dual substrates, one of which (D-mannitol) being specific for *P. veronii*, and the other (putrescine) for *P. putida*. Indeed, we could not measure growth of *P. putida* on D-mannitol, whereas *P. veronii* developed to one-fifth of its biomass on putrescine in comparison to D-mannitol, albeit with relatively slow growth (Fig 7A). However, in a 1:1 starting ratio co-culture on D-mannitol or putrescine, there was no distinguishable growth of *P. putida* or *P. veronii*, respectively, indicating effective unique and exclusive primary substrates to each of the species (Fig 7A). In contrast, the culture turbidity of the co-culture on putrescine was lower than that of *P. putida* alone on putrescine (Fig 7A). Co-cultures with both substrates simultaneously and with varying species' starting ratios (from 100:1 to 1:100, as before) showed growth of both species (judged from their specific fluorescence marker) to approximately the same final levels, independent of their starting ratio (Fig 7B). The apparent (visible) difference in onset of growth of either species in the co-culture is due to its varying population size in the starting mixtures, which is correctly simulated by the Monod growth model using fitted kinetic parameters in absence of any assumed cross-feeding (Fig 7C; the species-respective $\kappa_1$, $\kappa_2$ and $\kappa_3$ again deduced from the measured $\mu_{max}$, $K_S$ and yields as shown in Fig 2).

Although there was no competition for the primary added resources, we were interested to see whether population growth of both species would be independent of any appearing byproducts, in contrast to the case with succinate as a unique (competitive) resource. Indeed, both species grew in parallel in the co-cultures with both substrates (Fig 7C), with a mean stationary proportion of *P. veronii* across all starting ratios of 54.0% (± one SD of 3%) (Fig 7D). This is in contrast to the case of succinate, where the proportions of *P. veronii* varied between 0.2 and 60% (Fig 6E). In addition, one can observe how the community growth is almost a 'sum' of two overlaying independent species growth curves, which become more delayed, the smaller is the starting proportion of *P. veronii* (Fig 7C). The difference in the extension of the co-culture growth phase is due to the faster maximum growth rate of *P. putida* on putrescine (1.54 h $^{-1}$) than *P. veronii* on D-mannitol (0.36 h $^{-1}$) (S4 Fig).

Although the observations in this case were well explained by the growth kinetics alone of either species (i.e., without inclusion of cross-feeding), we still observed small deviations between the simulated (predicted) and observed stationary phase abundances of *P. veronii* (Fig 7D). In contrast to the case of succinate, it was the observed *P. veronii* stationary phase proportion that increased from 51.1% for a starting ratio of (*P. veronii*: *P. putida*) = 100: 1, to 57.6% for a starting ratio of 1:100, whereas the simulations predicted a constant ratio of 54%. Interestingly, the measured stationary phase abundance of *P. putida* cells remained the same among all starting ratios ($2.4 \pm 0.06 \times 10^9$ cells ml $^{-1}$, equivalent to a biomass of $1.69 \pm 0.05 \times 10^{-4}$ g mL $^{-1}$). However, a small temporal increase of *P. putida* biomass and fluorescence seemed to occur during the experiment (between 20 and 30 h) in co-culture with 100-fold starting excess of *P. veronii* (Fig 7B), suggesting an appearing cross-feeding. If at the same time the final measured cell number of *P. veronii* increased from $9.8 \times 10^8$ at a *P. veronii*: *P. putida* starting ratio of 100:1 to $1.4 \times 10^9$ cells per ml at a ratio of 1:100, this could mean that either the *P. veronii* cell size decreases or that more of the available resources is converted into biomass.

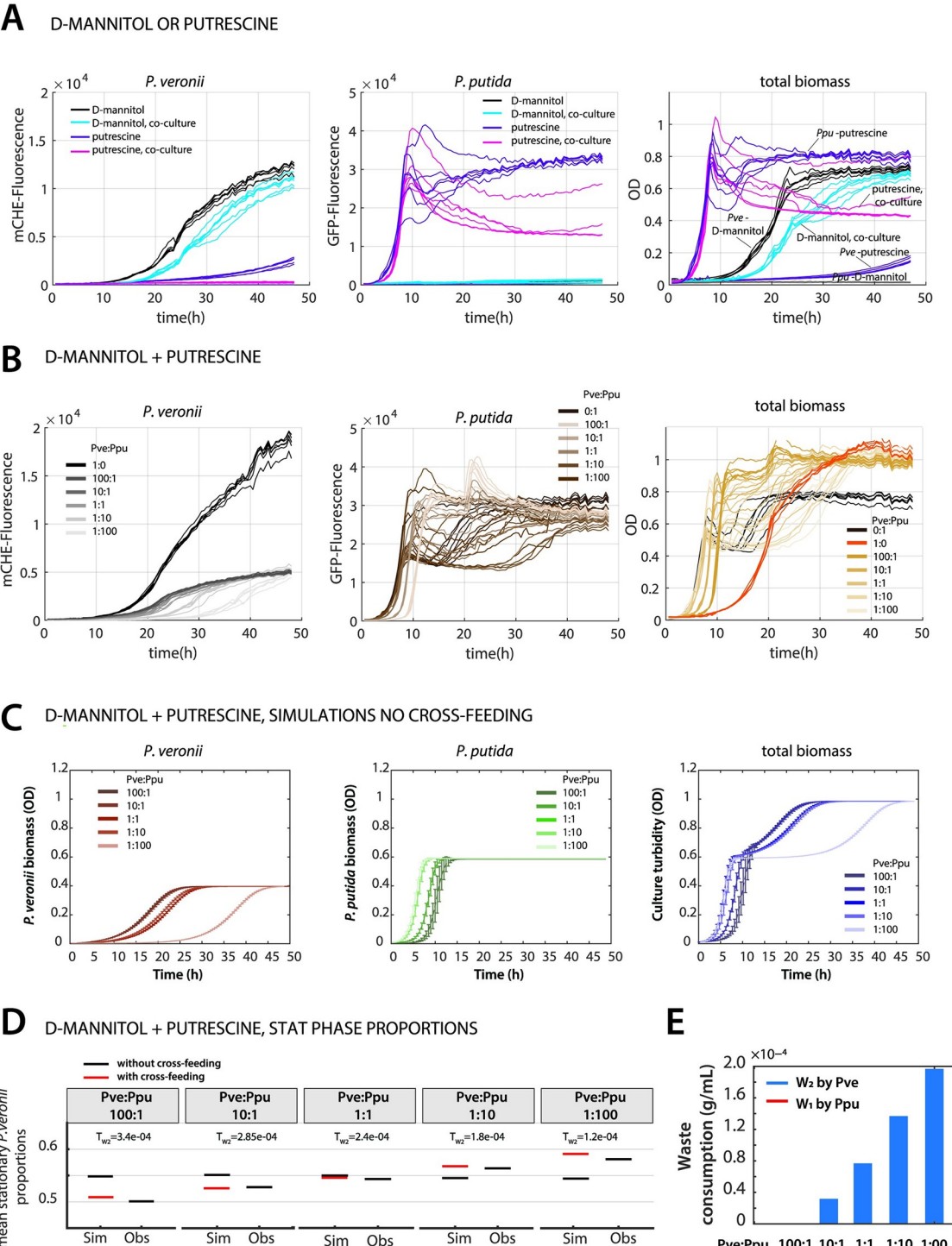

**Fig 7. Dual exclusive substrates in co-culture lead to independent growth of each species.** (**A**) Growth of *P. putida* or *P. veronii* alone and in 1:1 co-culture with either putrescine or D-mannitol. Graphs show development of mCherry fluorescence (specific for *P. veronii*), of GFP fluorescence (specific for *P. veronii*) and of culture turbidity (n = 6 replicates). (**B**) Growth of *P. putida* and *P. veronii* alone or in co-culture with mixture of 10 mM putrescine and 6.7 mM D-mannitol, at varying starting cell ratios (as indicated, n = 6 replicates). (**C**) Simulated growth of *P.veronii* and *P.putida*) under the same conditions and starting ratios as in B, using fitted kinetic parameters in absence of assumed cross-feeding. Data points show means from 8 replicates (points) ± their calculated standard deviation (bars). (**D**) Mean simulated (Sim) and measured (Obs, from flow cytometry) stationary phase proportions of *P. veronii* in the co-cultures, in absence (black) or presence (red) of assumed cross-feeding (discontinuous imposed threshold values

indicated). Variation of the data is smaller than the line size. (**E**) Calculated utilization of waste byproducts by *P. veronii* and *P. putida* for the threshold-imposed cross-feeding scenarios in panel D. S6 Data.

Despite both exclusive substrates, simulations thus again favoured the existence of cross-feeding, but in this case by *P.veronii* of byproducts coming from *P. putida*. Cross-feeding can again be described by a discontinuous threshold under constant *P. veronii* yield of 34.42% (Fig 7D). In contrast, the biomass of *P.putida* is not impacted by the cross-feeding, and its yield of 34% remains constant in models with and without cross-feeding. Allowing cross-feeding to occur would then lead to utilization of byproducts as visualized in Figs 7E and S5.

## 3 Discussion

We developed and verified a cross-feeding model for two-species interactions, which links appearing (summed) metabolites in the extracellular medium to changes in growth kinetic parameters. Cross-feeding was best explained with an activation threshold function, which could be experimentally parameterized under two interaction scenarios: (i) primary single substrate competition and (ii) substrate indifference. Comparisons of model simulations to experimental data showed that modelling with inherent Monod kinetic parameters without interspecific interaction terms are insufficient to explain co-culture growth. This was particularly true for cases where the initial abundances of both species were unequal, and the numerical minority could proliferate more than expected. From an ecological perspective, this is an important notion, because it would allow minority species to grow and survive better than expected from pair-wise competition assays that are conducted at equal species-ratios. We further showed that the model holds for both the situation of imposed primary substrate competition and the substrate indifference, albeit it with reversal of the species profiting most from the metabolite cross-feeding.

Our mathematical description extended generalized consumer-resource models [17] with interspecific interactions from chemical reaction network theory [20], which is similar to propositions as in Ref. [41] that assume cross-feeding mechanisms from metabolic generalists to other species on their metabolic by-products. Although these authors experimentally proved that such cross-feeding mechanisms are necessary to ensure observed species coexistence in microbial communities through collective interactions, and that metabolite concentrations are high enough to ensure growth of other species, their proposed permanent cross-feeding loops [41] cannot mathematically hold. As we demonstrate in Appendix D in S1 Text, Proposition 1; permanent cross-feeding loops would lead to vanishing equilibrium waste concentrations and biphasic co-culture growth (Fig 5B), which we did not observe empirically.

As an alternative, we introduced a regulatory mechanism that allows for activation of the cross-feeding positive feedback only above a threshold concentration, which overall gave better predictions of experimentally observed co-culture growth in a mean-field resource-limited environment (such as batch liquid suspended cell culture). The lumping of metabolites into a single waste is a simplification of approaches by Liao and coworkers [40], but facilitates the estimation of cross-feeding yield gains as shown here, under conditions where the nature of all metabolites is not known.

Regulated interaction networks have been previously used to describe gene regulation, where (positive and negative) feedback reactions can be active (ON) or inactive (OFF) depending on the concentration of transcription factors or inducers and effectors. For example, during transcription, which is inherently noisy, promoters switch randomly between ON and OFF states. The switching rate $\phi(F)$ depends in a nonlinear way on transcription factor concentration $F$, which is properly described by a Hill activation function $\phi$. The reader can

consult, e.g., [51] or [52, 50] for mathematical results on stochastic and deterministic systems involving such ON-OFF switching. We propose here that, similarly, activation functions can be used to describe interspecific growth interactions resulting from excreted/shared metabolites during primary growth.

The difficulty in modelling community growth under inclusion of cross-feeding interactions is that, foremost, maximum growth rates are not known for many of the constituting taxa; neither are their substrate dependencies nor their capacities and rates to utilise metabolites appearing during growth of other neighbouring taxa (e.g., as in [40]). To make more inclusive models, one could expand the simple 'summed' waste concentration as we proposed here, by matrices that would cover all (major) individual relevant substrates and products, with measured or estimated rate constants; for example, as proposed in [40, 41]. One could imagine that characterizing the single species substrate and metabolite 'landscape' will be facilitated in the future from detailed genome-scale metabolic models, which can also put boundaries on possible growth and reaction rates [13, 53, 54]. Alternatively, one could use lumped rate constants inferred from mono- and co-culture growth data, such as utilised here and proposed elsewhere [14]. Although we did not specifically measure metabolite concentrations appearing during mono- and co-culture growth, there is ample evidence for re-usable byproducts to appear and in the concentration ranges [45–49] that we infer here from kinetic parameter fitting (see, e.g., Fig 4). In addition, we estimate that approximately 25% of the yield in imbalanced starting ratios may originate from cross-feeding mechanisms.

We rigorously confronted and validated the model's predictions with experimental data for a two-species ecological network. Our experimental results were obtained in a suspended growth liquid environment, which leads to a mean field mathematical model. It will be interesting now to extend to controlled experiments with microbial communities and metabolic networks comprising more than two species, in order to assess the general usefulness of the regulated bacterial network models for predictions of community growth and its corresponding taxa composition. Furthermore, it is relatively straightforward to expand the mathematical framework to spatial situations with multiple species, growing at the expense of diffusible substrates, which will get us closer to understanding and predicting spatially structured communities, see e.g. [17].

## 4 Materials and methods

### 4.1 Bacteria cultivation

*P. veronii* 1YdBTEX2 (Pve, strain 5336) is a mini-Tn5 inserted tagged variant constitutively expressing GFP from the $P_{circ}$ promoter of ICEclc [55], [56]. *P. putida* F1 strain 5789 (Ppu) is a mini-Tn5 tagged variant of the wild-type strain [57] carrying a single-copy *mcherry* gene under control of $P_{tac}$. Tagging of F1 was accomplished by conjugation of the corresponding mini-Tn5 construct from *E. coli* DH5$\alpha$ − $\lambda$ pir with *E. coli* HB101 as helper for conjugation. Glycerol stocks of *P. veronii* and *P. putida* from −80˚C were plated on nutrient agar containing gentamycin at 10 mg L$^{-1}$ for selection of the mini-Tn5 constructs. Single freshly grown colonies were transferred to liquid 21C minimal media (MM) [58] in glass flasks with 5 mM succinate, and incubated at 30˚C at 180 rpm for 16–20 h. To prepare cells for competition experiments, culture samples (10 ml) were centrifuged at 5000 rpm in an F-34–6-38 rotor in a 5804R centrifuge (Eppendorf AG) for 10 min at room temperature. The supernatant was discarded, and the cell pellet was resuspended in 10 ml of minimal media salts (MMS, containing, per litre: 1 g NH$_4$Cl, 3.49 g Na$_2$HPO$_4$·2H$_2$O, 2.77 g KH$_2$PO$_4$, pH 6.8). The cell density in these suspensions was measured by flow cytometry (see below), after which the suspensions were used for the competition experiments.

Washed cultures were freshly diluted in liquid medium with 5 mM of succinate, or with 10 mM D-mannitol (for *P. veronii*), and 6.7 mM putrescine (for *P. putida*) as sole added carbon and energy source, either as individual monoculture or as binary mixture, each in ten replicates. Individual (mono-)cultures started with $10^6$ cells ml$^{-1}$. In the co-cultures, the initial cell ratio between *P. veronii* and *P. putida* was varied from 100:1, 10:1, 1:1, 1:10 and 1:100, each time starting with a mixture that contained $10^6$ cells ml$^{-1}$. In case of the substrate indifference experiment, we relied on diluting the starting cell suspensions to a culture turbidity at 600 nm (OD600) of 0.05 for *P. veronii* and 0.02 for *P. putida*. Cultures were incubated at 28–30˚C and growth was followed during 24–48 hours by automated reading of the culture turbidity and of GFP/mCherry fluorescence, every 15 (for succinate) or 30 min (for D-mannitol and putrescine) in a 96-well plate reader. After 24 (for succinate) and 48 h (for D-mannitol and putrescine), three replicates of each monoculture and of the co-culture were sampled, diluted and measured by flow cytometry to obtain the exact cell density. These measurements were used to determine a global conversion factor between the number of cells and the culture turbidity ('optical density', OD), while acknowledging that this doesn't take cell size changes or cell clumping into account. Measured culture densities were corrected for the optical density of the medium itself.

## 4.2 Flow cytometry

The cell density in the washed cultures was quantified either using a BD LSRFortessa (BD Biosciences, Allschwil, Switzerland) or a Novocyte flow cytometer (ACEA Biosciences, USA). For quantification using the BD LSRFortessa flow cytometer, an aliquot of 50 µl of bacterial culture was mixed with 2 mL sterile saline (9 g NaCl L$^{-1}$) solution in a 5 mL polystyrene tube (Falcon, Corning, NY, USA). The following parameter settings were used for cell quantification: FSC = default, SSC = default, FITC = 650 V, PE-Texas-Red = 650 V, injection flow rate = 35 µl min$^{-1}$, and acquisition time = 60 s. Four different FSC-H thresholds were tested for quantification: 400, 600, 800, and 1000. *P. veronii* was quantified on its FITC-signal (GFP), *P. putida* on the PE-Texas-Red signal (mCherry). For detection and quantification using the Novocyte flow cytometer (ACEA Biosciences, USA), washed liquid cultures were diluted 100–1000 times in MMS and 20 µl were aspired at 66 µl min$^{-1}$. *P. putida* cells were identified at an FSC-H threshold above 500, SSC-H above 150, and a PE-Texas Red-H signal gated to above an empty control (typically: 1000; channel voltage set at 592 V, representative for the mCherry fluorescence). *P. veronii* cells were identified at the same FSC-H and SSC-H thresholds but on the basis of an FITC-H signal above the empty control (typically, 600; representative for GFP fluorescence; channel voltage at 441 V).

## 4.3 Liquid culture competition experiments

To test competition between *P. veronii* and *P. putida* in liquid culture, we inoculated strains either individually or in combination into MM medium with either succinate (competitive scenario) or a combination of D-mannitol and putrescine as carbon substrates (indifference scenario). For succinate we used 5 ml MM medium with 10 mM succinate in a 29-ml sterile capped glass vial, with access to ambient air, and incubated cultures at 30˚C at 180 rpm. The cultures were sampled after 1, 24, 48, 72, and 144 h by removing a 50–µl aliquot, which was diluted with 2 ml of sterile water, to quantify the cell density of either *P. veronii* or *P. putida* using flow cytometry.

Yields from cell numbers and biomass were compared through measured cell volumes with succinate, D-mannitol or putrescine as substrates. The cell dimensions were quantified from phase-contrast microscopy images of culture samples using Fiji [59]. On succinate, the average

cell volume of *P. veronii* amounted to 1.8 μm$^3$ and that of *P. putida* to 0.8 μm$^3$. This was converted to a per cell carbon mass using the conversion factor of 0.264 pg C for an average *E. coli* cell volume of 2 μm$^3$ [60]. Cell and biomass yields were further inferred from weighed 70˚C-dried filtered samples from flow-cytometry counted stationary phase cultures grown on 10 mM succinate. For *P. veronii* this corresponded to 0.38 mg and $8.9 \times 10^8$ cells per OD; for *P. putida* to 0.46 mg and $9.5 \times 10^8$ cells per OD.

### 4.4 Two-species growth models

Growth in two-species communities was modelled from a framework similar to chemical reaction kinetics (CRN), describing substrate consumption, biomass formation and waste production in well-mixed environments (Appendix E in S1 Text, Eq. 29–31). Modelling with CRN allows the use of both stochastic and deterministic tools, as demonstrated by Ref [50]. In the case of the reciprocal waste utilization by two species under a single competitive limited substrate in well-mixed batch culture, with the activation function $\phi$, the resulting RBN looks like

$$
S_1 + R \xrightarrow{\kappa_1} 2S_1, \quad S_1 + R \xrightarrow{\tilde{\kappa}_1} S_1 + W_1,
$$
$$
S_2 + R \xrightarrow{\kappa_2} 2S_2, \quad S_2 + R \xrightarrow{\tilde{\kappa}_2} S_2 + W_2,
$$
(4)

$$
S_1 + W_2 \xrightarrow{\phi\left(F_2 - T_{W_2}\right)\kappa_{21}} 2S_1, \quad S_1 + W_2 \xrightarrow{\phi\left(F_2 - T_{W_2}\right)\tilde{\kappa}_{21}} S_1 + W_1,
$$
$$
S_2 + W_1 \xrightarrow{\phi\left(F_1 - T_{W_1}\right)\kappa_{12}} 2S_2, \quad S_2 + W_1 \xrightarrow{\phi\left(F_1 - T_{W_1}\right)\tilde{\kappa}_{12}} S_2 + W_2,
$$
(5)

with associated mass-action o.d.e.,

$$
\frac{dX_1(t)}{dt} = \kappa_1 X_1(t)C(t) + \phi(F_2 - T_{W_2})\kappa_{21}X_1(t)F_2(t),
$$
$$
\frac{dX_2(t)}{dt} = \kappa_2 X_2(t)C(t) + \phi(F_1 - T_{W_1})\kappa_{12}X_2(t)F_1(t),
$$
$$
\frac{dF_1(t)}{dt} = \tilde{\kappa}_1 X_1(t)C(t) - \phi(F_1 - T_{W_1})(\kappa_{12} + \tilde{\kappa}_{12})X_2(t)F_1(t) + \phi(F_2 - T_{W_2})\tilde{\kappa}_{21}X_1(t)F_2(t),
$$
(6)
$$
\frac{dF_2(t)}{dt} = \tilde{\kappa}_2 X_2(t)C(t) - \phi(F_2 - T_{W_2})(\kappa_{21} + \tilde{\kappa}_{21})X_1(t)F_2(t) + \phi(F_1 - T_{W_1})\tilde{\kappa}_{12}X_2(t)F_1(t),
$$
$$
\frac{dC(t)}{dt} = -(\kappa_1 + \tilde{\kappa}_1)X_1(t)C(t) - (\kappa_2 + \tilde{\kappa}_2)X_2(t)C(t).
$$

where $X_i$ are the biomass concentrations of species $S_i$, $i = 1, 2$; $F_k$, $k = 1, 2$ are the concentrations of waste $W_k$, $k = 1, 2$, and where $C$ is the primary resource concentration. In the used RBN model simulations, $\phi$ can take the form of $\phi \equiv 1$ (for constant waste utilization), $\phi(x) = \mathbb{1}_{\{x>0\}}$ (discontinuous cross-feeding), $\phi(F_i) = \dfrac{1}{1 + \left(\frac{T_{W_i}}{F_i}\right)^{k_i}}$ (for Hill activation function) or $\phi \equiv 0$ (no cross-feeding). Please refer to S1 Text for the full development of the mathematical proofs and details of the various other model scenarios.

### 4.5 Model parametrization

We assume that the $d$-dimensional o.d.e. which models species population growth is of the form

$$\frac{dX}{dt} = F(X, \theta), \tag{7}$$

where the unknown parameter $\theta \in \mathbb{R}^p$ contains the reaction rate constants of the underlying CRN. So, given a parameter $\Theta$, one can simulate the solution of the o.d.e. for some given initial condition to get $X(t, \theta)$, $t \geq 0$. Then, the value of the resulting solution is compared to the observed experimental values $y_n$, for time points $t_n$, $n = 1, \ldots, N$. The statistical model includes noise terms and is of the form

$$y_n = x_n(\theta) + \varepsilon_n, \tag{8}$$

where $x_n(\theta) = X(t_n, \theta)$ and the $\varepsilon_n$ model experimental errors. This process is replicated $R$ times, each simulation and experimental replicates involving $N_r$ time points. The resulting replicates are denoted by $y_{r,n}$, $x_{r,n}(\theta)$ and $\varepsilon_{r,n}$, $r = 1, \ldots, R$. We set for convenience $y_r = (y_{r,n})_{r=1,\ldots,N_r}$.

Reaction rates constants for *P. veronii* and *P. putida* were extracted from monoculture growth using the Metropolis-Hasting algorithm with Markov Chain Monte Carlo approach to estimate their probability distributions. We use the Adaptive Scaling Metropolis algorithm as described in Refs. ([61], [62]). The advantage of this procedure is that it estimates the variation in the growth rates, which can be included in simulations of the experimental results. For this, we separated the complete dataset (n = 10 replicates) in two halves. On one, we estimated growth rates and the remaining data were used to compare simulations and empirical observations.

All models and simulations were implemented in Matlab (vs.2020a, MathWorks Inc.).

### 4.6 Statistical procedures

Mean values for the different replicates in Fig 6E were compared by t-testing. Before performing the t-test, the assumptions of normality of the data and homogeneity of the variances were verified using the Shapiro-Wilk and Bartlett's tests, respectively.

## Supporting information

**S1 Fig. Co-culture representations.** (A) The predator-prey Lotka-Volterra representation where species $S_1$ preys on $R$ and $W_2$ and $S_2$ preys on $R$ and $W_1$. (B) The catalytic conversion representation involving species $R$, $W_1$ and $W_2$. The species associated to the dashed arrows indicate the nature of the catalyst, and the direction of the arrow indicates the transformation product.
(PDF)

**S2 Fig. Hill function examples of waste utilization.** Figure shows activation function outcome dependent of the waste concentration for different $k$-values of the Hill function, and in comparison to the indicative (step) function.
(PDF)

**S3 Fig. Effect of inhibitory interactions on the *P. veronii* and *P. putida* co-culture outcomes with succinate as sole shared competing substrate.** (A) Simulated versus empirical observed *P. veronii* stationary phase proportions at different starting cell ratios for the reciprocal and unilateral inhibition. (B) as A, but for the sum of co-culture biomass. (C) Effect on the ratio of

simulated versus observed *P. veronii* steady-state proportions for the model without any assumed interactions ($\phi = 0$, dark blue lines) and reciprocal interactions (red lines, n = 5 simulations). (D) as (C), but with unilateral inhibition. Used threshold values for the reciprocal inhibition are $T_{W_2} = 8.85e - 05$, $T_{W_1} = 2.19e - 04$ and Hill factor $k = 45$. For the unilateral inhibition simulation, we used $T_{W_1} = 9.03e - 04$ and a Hill factor of $k = 36$.
(PDF)

**S4 Fig. Estimated growth rates of *P. veronii* on D-mannitol and of *P. putida* on putrescine.** Plots show histograms of maximum growth rates inferred from Metropolis-Hasting fitting with the Markov Chain Monte Carlo approach, as described in the Materials and Methods section.
(PDF)

**S5 Fig. Simulated biomass growth and waste formation in co-cultures of *P. putida* and *P. veronii* under conditions of substrate indifference.** Plots show simulated biomass growth in 8 replicates of *P. putida* (blue) or *P. veronii* (green) in co-culture on a mixture of D-mannitol (red) and putrescine (brown), and predicted waste concentrations (light green and blue), for five different cell starting ratios (100:1, 10:1, 1:1, 1:10 and 1:100, as indicated), and with $1 \times 10^6$ cells per ml at start. Simulations in (A) assume a Monod model without cross-feeding, whereas (B) includes cross-feeding using the discontinuous threshold function).
(PDF)

**S1 Data. Source data for Fig 2.** *P. veronii* and *P. putida* extracted growth kinetic parameters.
(XLSX)

**S2 Data. Source data for Fig 3.** Growth of *P. veronii* and *P. putida* in mono- and cocultures on succinate.
(XLSX)

**S3 Data. Source data for Fig 4.** Effect of initial founder populations and cell ratios on competitive outcome of *P. veronii* and *P. putida* growth in cocultures on succinate.
(XLSX)

**S4 Data. Source data for Fig 5.** Simulated substrate utilization and byproduct formation in succinate cocultures of *P. veronii* and *P. putida* under different versions of the indicative function $\phi$.
(XLSX)

**S5 Data. Source data for Fig 6.** Effect of threshold parameter values on simulated steady state ratios of *P. veronii* and *P. putida* in cocultures on succinate.
(XLSX)

**S6 Data. Source data for Fig 7.** Observed and simulated growth of *P. veronii* and *P. putida* in mono- and cocultures on putrescine and/or D-mannitol.
(XLSX)

**S1 Text. Supporting information.** Appendix A: Logistic model for mono- and co-culture growth. Appendix B: Monod model for mono- and co-culture growth. Appendix C: Resource indifference. Appendix D: Vanishing waste concentration under permanent cross-feeding. Appendix E: Generalized consumer-resource model. Appendix F: RBN mass action kinetics. Appendix G: Inhibitory interactions. Appendix H: Metropolis-Hasting algorithm.
(PDF)

## Acknowledgments

The authors thank Nicolas Carraro for his help in part of the competition experiments and Noushin Hadadi for initial coding of a co-culture growth model.

## Author Contributions

**Conceptualization:** Isaline Guex, Christian Mazza, Manupriyam Dubey, Jan Roelof van der Meer.

**Data curation:** Christian Mazza, Manupriyam Dubey, Maxime Batsch, Renyi Li, Jan Roelof van der Meer.

**Formal analysis:** Isaline Guex, Christian Mazza, Renyi Li, Jan Roelof van der Meer.

**Funding acquisition:** Christian Mazza, Jan Roelof van der Meer.

**Investigation:** Isaline Guex, Christian Mazza, Manupriyam Dubey, Maxime Batsch, Renyi Li, Jan Roelof van der Meer.

**Methodology:** Isaline Guex, Christian Mazza, Manupriyam Dubey, Maxime Batsch, Renyi Li, Jan Roelof van der Meer.

**Project administration:** Christian Mazza, Jan Roelof van der Meer.

**Software:** Isaline Guex.

**Supervision:** Christian Mazza, Jan Roelof van der Meer.

**Validation:** Isaline Guex, Christian Mazza.

**Visualization:** Isaline Guex, Jan Roelof van der Meer.

**Writing – original draft:** Isaline Guex, Christian Mazza, Jan Roelof van der Meer.

**Writing – review & editing:** Isaline Guex, Christian Mazza, Manupriyam Dubey, Maxime Batsch, Renyi Li, Jan Roelof van der Meer.

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
