## [Decision Letter · Decision Letter 0]

8 Jun 2023

Dear Prof. van der Meer,

Thank you very much for submitting your manuscript "Regulated bacterial interaction networks: A mathematical framework to describe competitive growth under inclusion of metabolite cross-feeding" for consideration at PLOS Computational Biology.

As with all papers reviewed by the journal, your manuscript was reviewed by members of the editorial board and by several independent reviewers. In light of the reviews (below this email), we would like to invite the resubmission of a significantly-revised version that takes into account the reviewers' comments.

We cannot make any decision about publication until we have seen the revised manuscript and your response to the reviewers' comments. Your revised manuscript is also likely to be sent to reviewers for further evaluation.

Sincerely,

Pedro Mendes, PhD

Section Editor

PLOS Computational Biology

Natalia Komarova

Section Editor

PLOS Computational Biology

Reviewer's Responses to Questions

**Comments to the Authors:**

Reviewer #1: In this manuscript, Guex et al. developed a metabolite-based cross-feeding model and used it to explain the population dynamics of a two-species coculture system. By evaluating different model assumptions, they found that a concentration-dependent threshold function for byproduct utilization can best explain the observed data. They concluded that cross-feeding enables minority species to proliferate better when cross-feeding is allowed, compared to cross-feeding-free models.

First, the description of model development in this manuscript appears unnecessarily lengthy. The first section of Results (section 3.1) are technical, and it would be helpful to move these details to the Methods or Supplementary Material. I appreciate the authors’ approach of introducing their model in a stepwise manner, starting from a single species model all the way up to a 2-species model with regulated cross-feeding. However, most of these models have been established. Therefore, a concise summary of the key differences between the authors’ model and past models would suffice for readers to understand the innovation.

Following my previous comment, the model description in the Methods section (5.2) is not helpful too. Introducing the basic concepts of CRN models, mass action kinetics, and generalized consumer resource models seems to be unnecessary, as they are quite standard. If the authors still want to include them, the Supplementary Material would be a better place. Instead, a clear description of the equations used in the CURRENT study is expected in the Methods section. Although the authors provide an example in Section 5.2.3, this is poorly explained. For example, the waste concentration F3 is undefined since there are only two waste products in CRN. It is also unclear whether kappa_i and kappa_i_tilde are fundamental parameters or functions of other fundamental parameters. Furthermore, it is not clear whether the example was a generalized consumer-resource model or a model developed by this study.

Based on the explanation provided, I am concerned that the model presented in this study is almost identical to, and even simpler (e.g., no metabolite-dependent growth inhibition) than, the model developed in Liao et al. PLoS Comput Biol , 2020 (https://journals.plos.org/ploscompbiol/article?id=10.1371/journal.pcbi.1008135). Both are kinetic, metabolite-based cross-feeding models, with the potential difference being that the current study used two activation functions to simulate the uptake rate of metabolic byproducts. However, the underlying math of the two functions does not make sense to me. The authors called Eq. 6 “permanent cross-feeding” and Eq. 9 “regulatory cross-feeding”. For Eq. 6, the reaction rate under mass action kinetics would be kappa_14*[S_1]*[W_2], which depends on the byproduct concentration [W_2]. Since this term goes to zero when [W_2] equals zero, the cross-feeding rate it quantifies cannot be permanent. Moreover, the mass action kinetics of Eq. 6 (i.e., kappa_14*[S_1]*[W_2]) already accounts for the concentration-dependent byproduct uptake. Why did the authors introduce another concentration-dependent functions (Eq. 9 and 10) that models the same mechanism? I would assume the reaction rate of Eq. 9 is phi(F_2-T_w1)*kappa_14*[S_1]*[W_2](=F_2). If this is correct, both phi(F_2-T_w1) and F_2 quantify the concentration-dependent effect. At the minimum, the authors need to compare their model with the one presented in Liao et al and justify why they introduced two additional regulatory functions given that the concentration-dependency of byproduct uptake can be modeled by mass action kinetics of Eq. 6.

Aside from the mathematical issues, my other concern is with the conclusion that cross-feeding is responsible for the difference between the observed and simulated species ratios in a “cross-feeding-free” model. While cross-feeding is a possible explanation for this difference, many other mechanisms can be explanatory too, especially considering that the model-experiment difference is generally small (though significant) across all experimental settings (Fig. 4C). For example, the model did not account for the inhibitory effects of byproduct wastes or the proportion of wastes that could be utilized by the other microbe (all wastes were combined into a pseudo molecule). Is it feasible for the authors to validate the presence of cross-feeding and determine the specific metabolites that mediate this positive interaction?

Other specific comments:

The authors used “positive feedback loop” in a couple of places. Cross-feeding is for sure a positive interaction (a -> b) but I am not sure if it forms an overall loop (a -> b -> a).

The notations used in Eq. 9 are a bit confusing. Why does the uptake of byproduct W_2 have a threshold of T_w1 while the uptake of byproduct W_1 has a threshold of T_w2? Is this a typo in the subscripts?

A parenthesis is missing somewhere between line 238-241

Line 283-284: could not find model B of equation 7

Reviewer #2: See the attached review report

**Have the authors made all data and (if applicable) computational code underlying the findings in their manuscript fully available?**

Reviewer #1: **No: **Scripts used for the models in this work can be accessed from https://github.com/IsalineLucille22/Liquid-models.git. However, this link is not accessible yet and will be made public only upon acceptance.

Reviewer #2: Yes

PLOS authors have the option to publish the peer review history of their article (what does this mean?). If published, this will include your full peer review and any attached files.

Reviewer #1: No

Reviewer #2: No
---

## [Decision Letter · Decision Letter 1]

31 Jul 2023

Dear Prof. van der Meer,

We are pleased to inform you that your manuscript 'Regulated bacterial interaction networks: A mathematical framework to describe competitive growth under inclusion of metabolite cross-feeding' has been provisionally accepted for publication in PLOS Computational Biology.

Best regards,

Pedro Mendes, PhD

Section Editor

PLOS Computational Biology

Natalia Komarova

Section Editor

PLOS Computational Biology

Reviewer's Responses to Questions

**Comments to the Authors:**

Reviewer #1: I appreciate the efforts made by the authors in addressing my questions. The technical inquiries have been satisfactorily addressed, though I believe there is still room for improvement in the writing. Nevertheless, great job and congratulations!

Reviewer #2: See the attached report

**Have the authors made all data and (if applicable) computational code underlying the findings in their manuscript fully available?**

Reviewer #1: Yes

Reviewer #2: None

PLOS authors have the option to publish the peer review history of their article (what does this mean?). If published, this will include your full peer review and any attached files.

Reviewer #1: No

Reviewer #2: **Yes: **Hao Wang

---

## [Editor Report · Acceptance letter]

16 Aug 2023

PCOMPBIOL-D-23-00220R1 

Regulated bacterial interaction networks: A mathematical framework to describe competitive growth under inclusion of metabolite cross-feeding

Dear Dr van der Meer,

I am pleased to inform you that your manuscript has been formally accepted for publication in PLOS Computational Biology. Your manuscript is now with our production department and you will be notified of the publication date in due course.

With kind regards,

Zsofia Freund
